# Identification of Video Game Addiction Using Heart-Rate Variability Parameters

**DOI:** 10.3390/s21144683

**Published:** 2021-07-08

**Authors:** Jung-Yong Kim, Hea-Sol Kim, Dong-Joon Kim, Sung-Kyun Im, Mi-Sook Kim

**Affiliations:** 1Department of HCI, Hanyang University ERICA, Ansan-si 15588, Gyeonggi-do, Korea; jungkim@hanyang.ac.kr (J.-Y.K.); kimheasol@hanyang.ac.kr (H.-S.K.); 2Department of Industrial and Management Engineering, Hanyang University ERICA, Ansan-si 15588, Gyeonggi-do, Korea; skalon87@gmail.com; 3Department of Clothing and Textiles, Kyung Hee University, Seoul 02447, Korea; mskim@khu.ac.kr

**Keywords:** HRV parameter, game addiction, *League of Legends*, stress response, sensitivity, specificity, logistic regression

## Abstract

The purpose of this study is to determine heart rate variability (HRV) parameters that can quantitatively characterize game addiction by using electrocardiograms (ECGs). 23 subjects were classified into two groups prior to the experiment, 11 game-addicted subjects, and 12 non-addicted subjects, using questionnaires (CIUS and IAT). Various HRV parameters were tested to identify the addicted subject. The subjects played the *League of Legends* game for 30–40 min. The experimenter measured ECG during the game at various window sizes and specific events. Moreover, correlation and factor analyses were used to find the most effective parameters. A logistic regression equation was formed to calculate the accuracy in diagnosing addicted and non-addicted subjects. The most accurate set of parameters was found to be pNNI20, RMSSD, and LF in the 30 s after the “being killed” event. The logistic regression analysis provided an accuracy of 69.3% to 70.3%. AUC values in this study ranged from 0.654 to 0.677. This study can be noted as an exploratory step in the quantification of game addiction based on the stress response that could be used as an objective diagnostic method in the future.

## 1. Introduction

The game industry is growing, with a market size of more than US $123.4 billion worldwide. South Korea is ranked fifth in the world, with 6.7% of the world market share [1], and accounts for 55.8% of Korea’s content industry exports in 2018 [2]. Ryu and Lee [3] stated that such booming of the game industry has a positive influence on society, including stress management, the realization of the ideal self, and physical ability improvement. In particular, in the current COVID-19 environment, online games are recognized as a complementary means of social distancing [4,5]. However, Internet game players are not protected from becoming addicted to gaming. This addiction problem could adversely affect personal life as well as family and society, and has become a serious public health issue. Byun and Lee [6] found that Internet addiction is closely related to the increased frequency and duration of Internet use, and leads to anxiety, fear, depression, and obsessive-compulsive disorder, with adolescents being vulnerable target users. Koepp et al. [7] observed that dopamine is secreted from the brains of addicted adolescents with a similar pattern to that of drug addiction.

Adverse effects on adolescents have been studied by many authors [8,9,10]. In particular, it is notable that the most influential factor causing Internet addiction is stress due to excessive competition, and that adolescents exposed to excessive stress sources were readily immersed in the Internet [11]. Adolescents often experience alienation or loneliness when they are addicted to Internet games [12]. They relieve the stress related to daily life and loneliness by using internet games, which were easily accessible [13]. The higher the level of stress, the more they tended to fall into game addiction [14]. According to a study by Lee [15], game addiction prevents adolescents from coping with stress sources properly, causing various psychological problems and stress responses. Likewise, the literature indicates that Internet game addiction and mental stress are closely related.

In recent years, heart rate variability (HRV) has been used in many studies to evaluate stress levels [16,17,18,19]. Since stress affects the autonomic nervous system (ANS), HRV controlled by the ANS is often referenced as a stress indicator. A number of studies on HRV parameters have been conducted in this regard. Taelman et al. [20] and Vuksanović and Gal [21] observed that the mean of the NN interval, which is often expressed as the RR interval, and the standard deviation of all NN intervals (SDNN) decreased significantly under mental stress. Taelman et al. [20] and Tharion et al. [22] showed that pNN50 (percentage of successive RR intervals greater than 50 ms) is significantly decreased under stress. Papousek et al. [23] and Traina et al. [24] reported an increase in the low-frequency power range (LF), a decrease in the high-frequency power range (HF), and a significant increase in the LF/HF ratio when subjects experience stress. Park et al. [25] tested the newly developed measuring system to examine electrocardiograms (ECGs) and found a consistent increase in HR and SDNN as the level of addiction increased. At the same time, the LF and LF/HF parameters showed an obvious increasing trend at a high level of addiction.

On the other hand, Hafeez et al. [26] used EEGs to classify game addicts and non-addicts using cluster analysis and pattern discrimination. They introduced a statistical method to quantify the addiction phenomenon, and Hafeez et al. [27] and Kim et al. [28] identified the theta and theta/alpha parameters of the right occipital region as the discriminating variables between addicts and non-addicts. Likewise, the attempt to quantify the particular characteristics of addiction is an ongoing topic for researchers. If such a numerically quantifiable approach can be successful and assist physicians in identifying an addicted patient, they will be able to treat the patient more efficiently and objectively. Therefore, in this study, the authors are challenged to search for a quantifiable indicator of addiction in ECG response by investigating various HRV parameters. The purpose of this study is to extract quantitative HRV parameters that characterize the particular stress response of game addicts. To achieve this research goal, an exhaustive approach was performed by testing all the candidate parameters collected using window sizes of 30, 60, 90, and 120 s.

## 2. Methods

### 2.1. Subjects

A total of 23 male students participated in the experiment. The mean age was 23 years (±3 years). Eleven participants were addicted, and 12 of them were non-addicted. They were categorized using the Compulsive Internet Use Scale (CIUS) by Meerkerk et al. [29], and the Internet Addiction Test (IAT) by Young and De Abreu [30]. Based on CIUS, subjects with 2.5 or higher were categorized as addicted, and those with scores less than 1.5 were categorized as non-addicted [31]. An IAT score of 50 or higher has been used to classify the game-addicted by many researchers [10,32,33,34,35]. In this study, a subject was categorized as a game addict only when the subject met both the IAT and CIUS standards. For non-addicted subjects, an IAT score of 40 or lower was required. 14 addicted subjects and 14 non-addicted subjects were selected. 3 addicted subjects and 2 non-addicted subjects were discarded due to a technical error in the measurement system. Controlling the compounding effect of gender in this study, only male participants were tested in this study. Alcohol consumption was prohibited for 24 h before the start of the experiment, and smoking and coffee consumption were prohibited for 1 h before the start of the experiment. A fee was paid to the participants. The experiment was conducted in accordance with the regulations under consideration by the Institutional Review Board of Hanyang University in the Republic of Korea (IRB approval number: HYU-2019-08-004-1).

### 2.2. Apparatus

The questionnaires used to categorize subjects into two groups prior to the experiment were the CIUS by Meerkerk et al. [29] and the IAT by Young and De Abreu [30,36].

*League of Legends* by Riot Games Inc. (Los Angeles, CA, USA) was chosen for the experiment. This game was one of the most frequently played games among internet game players [37], and the frequent battles in the game made players experience a simulated life and death situation associated with probable stress reactions.

For data collection, an auxiliary channel of QEEG-64FX by LAXTHA Inc. (Daejeon, Republic of Korea) was used for ECG measurements (Figure 1). A data collection program called Telescan was used. The data sampling rate was set to 500 Hz. The experiment was conducted in a room equipped with a computer, a table, and a chair, where other external stimuli were restrained.

### 2.3. Experimental Design

The experiment was designed to test HRV parameters to determine whether they could differentiate subjects into two groups: addicted and non-addicted. A between-subjects design was used in this study. The independent variables were the addiction status of the group, and the dependent variables were 14 parameters, including 7 time-domain variables and 7 frequency-domain variables. The time-domain parameters are NN interval average (RR interval average), SDNN, SDSD, pNNI50, pNNI20, RMSSD, and heart rate average (Table 1). The frequency-domain parameters are LF, HF, LF/HF ratio, LFnu, HFnu, total power, and VLF (Table 2). This study observed specific events during gameplay, including a “killed event”, when a player’s character was killed by an opponent, and a “killing event”, when the player killed an opponent. The data collection window sizes for these events were 30 s, 60 s, 90 s, and 120 s, respectively, to consider the possible delay of the response.

### 2.4. Procedure

Positive electrode was placed in the V1 location (between the right rib 3 and 4), and the negative electrode was placed in the left infraclavicular fossa according to the standard limb guidance method [39]. The experimental procedure was briefly explained to the subject, and the ECG sensors were attached and tested to ensure that stable signals were obtained for 1 min while the subjects were relaxing. A “normal game”, which is a practice game that does not affect the player’s score, was played for familiarization; a “ranked game”, which is a competing game affecting the player’s score, was played for 30–40 min. For players’ immersion in the game, the ranked game was played based on the individual skill level. Subjects played a “normal game” once and a “ranked game” twice, while the ECG was obtained. Subjects were not informed about the addiction test score; thus, they did not know whether they were categorized in the addicted group or not. The detailed experimental procedure is shown in Figure 2.

### 2.5. Data Analysis

The data were analyzed in batches using Python, and time series analysis and frequency analysis were performed at the same time. The parameters used for time series analysis were extracted by using the Christov ECG R-peak segmentation algorithm. The extracted parameters were NN interval average, SDNN, RMSSD, pNNI50, pNNI20, SDSD, and heart rate average. The signal was also extracted and transformed into frequency parameters using the fast Fourier transform. Welch’s periodogram was applied to estimate the spectral properties of the HRV signals, using a Hanning window. VLF (power in very-low-frequency ranges, 0.0033–0.04 Hz), LF (power in low-frequency ranges, 0.04–0.15 Hz), HF (Power in high-frequency ranges, 0.15–0.4 Hz), and total power (Power in all the frequency ranges, ≤0.4) were obtained by the sum of the power in the relevant frequency range of the spectrum. Based on these power values, the values of LF/HF ratio, LFnu, and HFnu were calculated.

Normality was tested by using Kolmogorov–Smirnov test for individual data set. The dataset with a low normality value was graphically examined to ensure an adequate level of normality. During the process, illegal outliers were treated. The *t*-test was performed (*p* < 0.1) to find the parameters and window size that statistically differentiate two groups: the addicted and non-addicted. The statistical analysis was an exhaustive process used to identify the set of most effective parameters and the window size. A correlation analysis was also performed to determine the redundancy of parameters, and a factor analysis was performed to choose the main parameters representing the characteristics of each group. Finally, a logistic regression analysis was conducted to test the sensitivity and specificity of the statistical model in identifying addicted or non-addicted subjects based on the current experimental data. The analysis process is illustrated in Figure 3. Statistical analysis was performed using SPSS Statistics 24.

## 3. Results

An elimination process was used to sort out the best combination of parameters out of 14 parameters from 4 window sizes through statistical analyses.

### 3.1. The t-Test Results between Groups by Window Size

There were no significant differences in average parameter values between the addicted and non-addicted groups for the entire window sizes during the experiment (*p* > 0.1).

### 3.2. The t-Test Results between Groups after Specific Event

There was no significant difference of HRV parameters between groups for window sizes of 30 s, 60 s, 90 s, and 120 s after “killing events” (*p* > 0.1). However, as shown in Table 3 and Table 4, the HRV parameters measured for window sizes of 30 s and 60 s after “killed events” showed a significant difference in some parameters between the two groups. In particular, pNNI20 and LF showed a significant difference (*p* < 0.05), and a marginally significant difference was observed for SDSD, RMSSD, and total power (*p* < 0.1).

### 3.3. Correlation Analysis and Factor Analysis with HRV Parameters

A correlation analysis was performed to examine the redundancy of the parameters in differentiating between the two groups. LF and pNNI20 with significant *p*-values (*p* < 0.05) in the *t*-test indicated a low correlation coefficient (0.264). Both could be used to improve statistical power in differentiating the two groups. On the other hand, SDSD and RMSSD showed a correlation coefficient of 1.000, and the total power and LF indicated a coefficient of 0.958. Thus, only one parameter was used to build the statistical model. Therefore, the correlation analysis suggested that the combination of the [pNNI20, LF, SDSD] or [pNNI20, LF, RMSSD] parameter set could be the best combination of parameters with the least redundancy.

Factor analysis was also performed to examine whether the selected parameters covered various factors of the data (Figure 4). The parameters with high eigenvalues for Factor 1 were RMSSD, SDSD, pNNI_50, and pNNI_20, and the parameters with high eigenvalues for Factor 2 were LF, total power, and SDNN. That is, the [pNNI20, LF, SDSD] or [pNNI20, LF, RMSSD] parameter set from the correlation analysis (Table 5) were found to have the highest eigenvalues for both Factor 1 and Factor 2 (Table 6). Therefore, the final combination of parameters for statistical modeling was [pNNI20, LF, RMSSD] or [pNNI20, LF, SDSD]. In logistic regression modeling, [pNNI20, LF, RMSSD] was arbitrarily selected to test the model performance in this study because both RMSSD and SDSD were highly correlated with each other (r = 1.000).

### 3.4. Logistic Regression Models

Logistic regression models were developed using the selected parameters. A total of 15 mathematical equations were designed to test the maximum sensitivity and specificity of the parameters using natural logarithms and squares. In terms of identifying the addicted group, the sensitivity was computed, and ranged from 0.324 to 0.400; the specificity ranged from 0.828 to 0.922. The overall accuracy ranged from 67.7% to 70.3%. The model with the highest specificity of 0.922 was constructed using pNNI20, ln(RMSSD), and LF. The model with the highest sensitivity of 0.400 was obtained using ln(pNNI20), (RMSSD)^2^, and ln(LF). The model with the highest overall accuracy of 70.3% was obtained using pNNI20, ln(RMSSD), and LF. The second-highest overall accuracy model (69.7%) was obtained using ln(pNNI20), ln(RMSSD), and (LF)^2^. The results are summarized in Table 7.

### 3.5. Characteristics of Distributions Affecting the Sensitivity and Specificity

The true positive rate (sensitivity) was less than 0.4 in the above analysis, which is not good enough to provide a diagnosis of addiction for medical treatment. Such a relatively low sensitivity could be a part of the outcome based on the logistic regression model to maximize the total accuracy. To see the characteristics of the probability distribution of the data, Figure 5, Figure 6 and Figure 7 are shown under the assumption of a normal distribution. As shown, there is a substantial overlap between distributions that could make either sensitivity or specificity low. From the observations, the criterion beta used for decision-making seemed to be biased to a conservative standard rather than a liberal one, considering that the specificity was much higher than the sensitivity. For example, for Model 1, with a maximum accuracy of 72.3%, the cut-off point associated beta value was set to 0.523, and the sensitivity and specificity were computed as 0.324 and 0.953, respectively. If a different cutoff value was then used, such as 0.372 in Model 2, the sensitivity and specificity can be computed as 0.656 and 0.703, respectively, with 67.3% accuracy.

### 3.6. Area under the Curve (AUC) Values

Figure 8 shows the ROC curves of the four models. The AUC value of 0.677 was for Model 1, 0.655 for Model 2, 0.673 for Model 3, and 0.654 for Model 4. According to Hosmer and Lemeshow’s study [40], models having an AUC value of 0.5 or less have no discriminating power. A model can be considered acceptable only if the AUC value is between 0.7 and 0.8, and a model has excellent discriminating power if the AUC value is between 0.8 and 0.9. Thus, the AUC value of the current logistic regression model is close to the acceptable level, but further refinement is required for the model to be acceptable.

## 4. Discussion

The study showed that “being killed” in a virtual situation generated a greater signal response among the addicted subjects than non-addicted subjects. Klimt et al. [41] mentioned that a shift in self-perception would occur while enjoying the game and identifying oneself with the game character or when playing games experiencing flow or psychological mastery. Turkay and Kinzer [42] stated that the customization process of avatars by players could greatly influence players to identify themselves as game characters. Therefore, it is reasonable to think that such an affective attachment with an avatar could psychologically influence the players, and this phenomenon could be even more severe among addicted subjects than non-addicted ones.

Regarding the model building, three different statistical methods were used to select the parameters to build the best logistic regression model. Through the *t*-test, the pNNI20 and LF parameters were selected because they showed the most significant results (*p* < 0.05) in differentiating the two groups 30 s after the “being killed” event. This means that both time-domain and frequency-domain parameters could be effective in statistically discriminating between the two groups. The total power parameter showed a significant *p*-value (<0.072); however, it was not selected for the final logistic model because it was highly correlated with the LF parameter (r = 0.958) to avoid redundancy. In addition, the RMSSD (or SDSD) parameter was used for the logistic regression model because it showed the highest eigenvalue (0.86) of Factor 1 in the factor analysis. The LF parameter with a significant *p*-value in the *t*-test also showed the highest eigenvalue (0.715) for Factor 2, which was used for the final logistic regression model.

The final parameters selected in this study were found to be associated with the stress response based on previous studies. Bernardi et al. [43] evaluated HRV parameters under the mentally stressful situation of a subject performing arithmetic while speaking or reading, and they observed the increased power of LF when subjects were hurrying to perform the calculation task. Huang et al. [44] found that RMSSD and the combination of various variables had a positive correlation with mental fatigue induced by mental stress. According to a study by Jang et al. [45], RMSSD was also found to have a marginal correlation with tension (r = 0.268, *p* = 0.039), depression (r = 0.356, *p* = 0.005), fatigue (r = 0.259, *p* = 0.041), and frustration (r = 0.304, *p* = 0.018). Lee et al. [46] observed changes in HRV during physical and mental stress in patients with depression, and they reported a significant increase in RMSSD during the stress period compared with the rest period. Mallinani et al. [47] explained that increased sympathetic activity could be functionally characterized by an increase in the LF component in terms of LF–HF balance. Kim et al. [48] reviewed the function of HRV parameters and concluded that low parasympathetic activity was frequently related to a decrease in HF and an increase in LF.

To investigate the efficacy of the regression model in diagnosing game addiction patients, the AUC values were calculated and compared with the reference values. The computed AUC value in this study ranged from 0.654 to 0.677, which is known to have insufficient accuracy for field applications. This indicated that the increased stress response of the addicted during a “killed event” was statistically meaningful, but it might not fully reflect the symptom of addiction that the players were experiencing. Regarding the sensitivity and specificity score, the sensitivity was computed and ranged from 0.324 to 0.400, and the specificity ranged from 0.828 to 0.922 based on the logistic regression model with the default cut-off point used as a decision criterion. However, the values could change when different cut-off points were used. For now, the AUC value was less than 0.7, which could expect only less-than-accurate decision-making. Therefore, it is necessary to test the model performance under various experimental conditions. At any rate, it is important to understand the nature of HRV parameters among addicted game players, who have been very responsive to stressful stimuli, which was worthwhile to investigate further for quantification of addictive symptoms during game playing.

## 5. Conclusions

In this study, the difference in HRV parameters between the addicted and non-addicted group was measured during game playing, and it was found that pNNI20, RMSSD, and LF reflected the difference in stress response sensitively for a window size of 30 s after a “being killed” event. To identify the difference between the game-addicted and non-addicted subjects, the AUC score was computed and found to be less than accurate. The quantification of the psychophysiological response of the addictive game was a challenging task, as was shown in this study, but it is worth pursuing the prevention and rehabilitation of addicted patients in the future. For further study, various types and greater numbers of subjects need to be tested for better representation of the addiction symptoms. Additional mathematical exploration using artificial intelligence techniques could be another option for analyzing bio-information with a high level of variability and probable irregularity. It would also be intriguing to examine and compare the HRV parameters to other psychophysiological signals to identify the unknown patterns of game addiction.

## Figures and Tables

**Figure 1 sensors-21-04683-f001:**
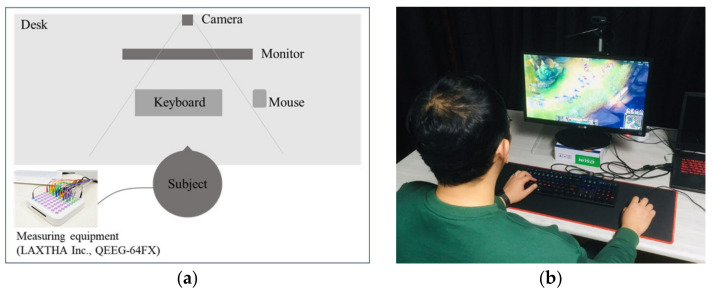
ECG measurement equipment. (**a**) Top view of experimental set-up, (**b**) The experimental scene.

**Figure 2 sensors-21-04683-f002:**
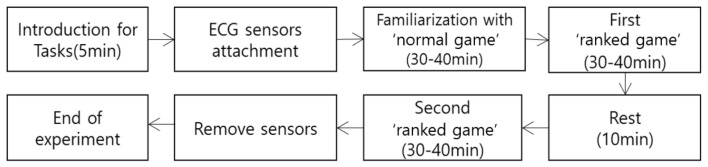
Experimental process.

**Figure 3 sensors-21-04683-f003:**
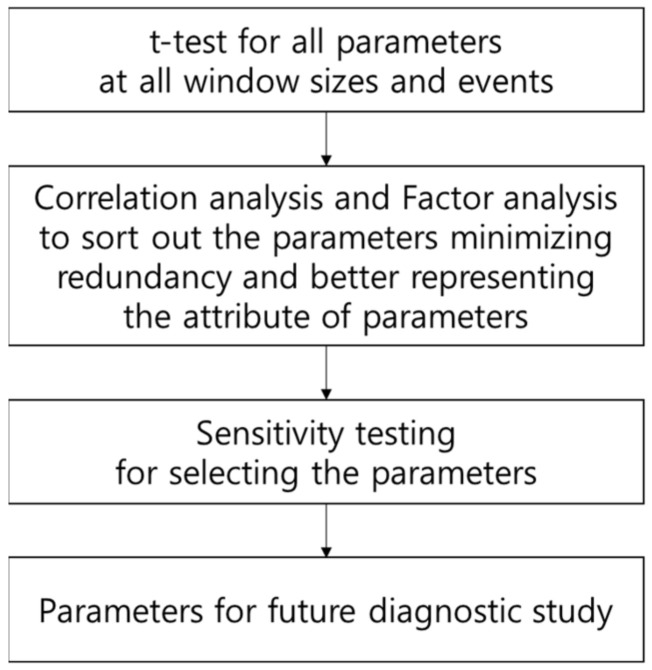
Data analysis process.

**Figure 4 sensors-21-04683-f004:**
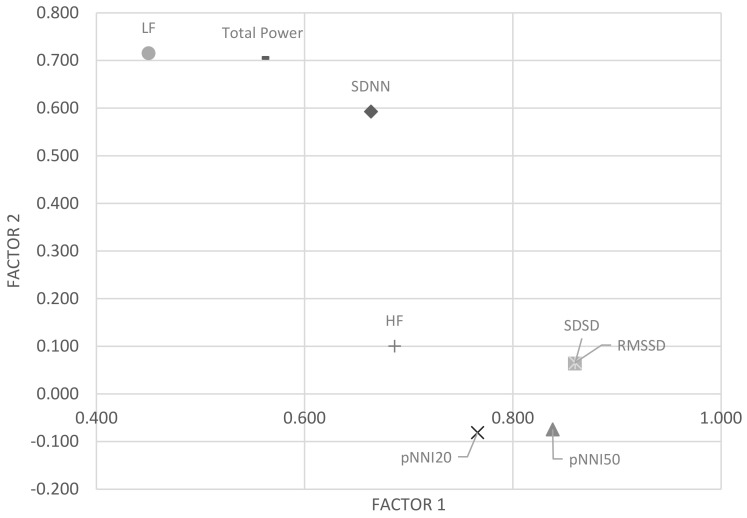
Factor analysis results.

**Figure 5 sensors-21-04683-f005:**
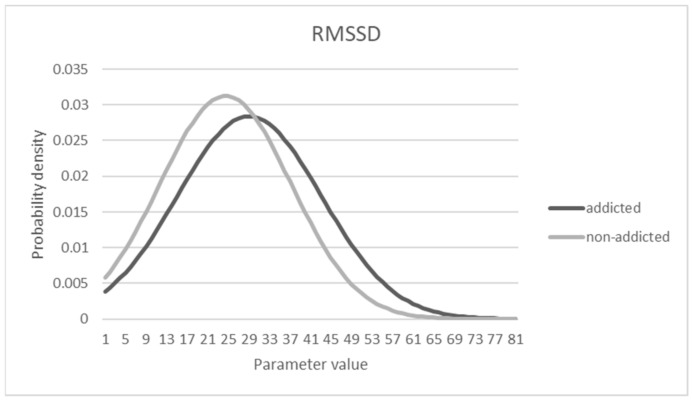
Probability density distribution of RMSSD parameter.

**Figure 6 sensors-21-04683-f006:**
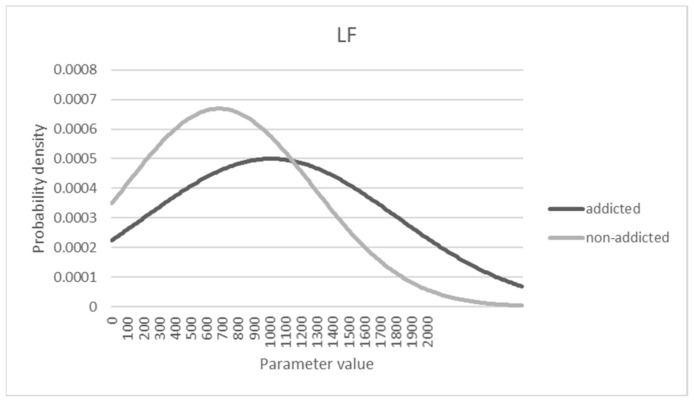
Probability density distribution of LF parameter.

**Figure 7 sensors-21-04683-f007:**
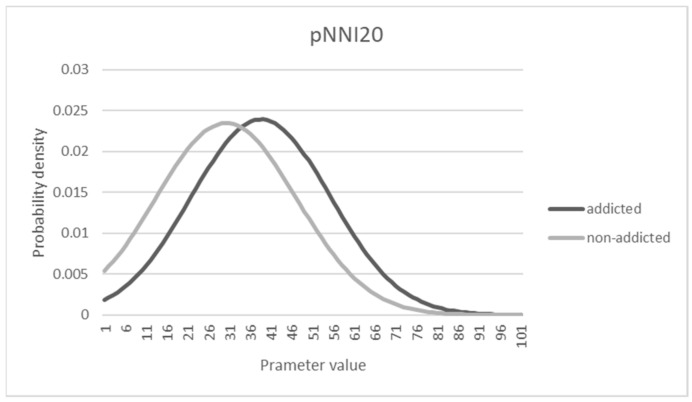
Probability density distribution of pNNI20 parameter.

**Figure 8 sensors-21-04683-f008:**
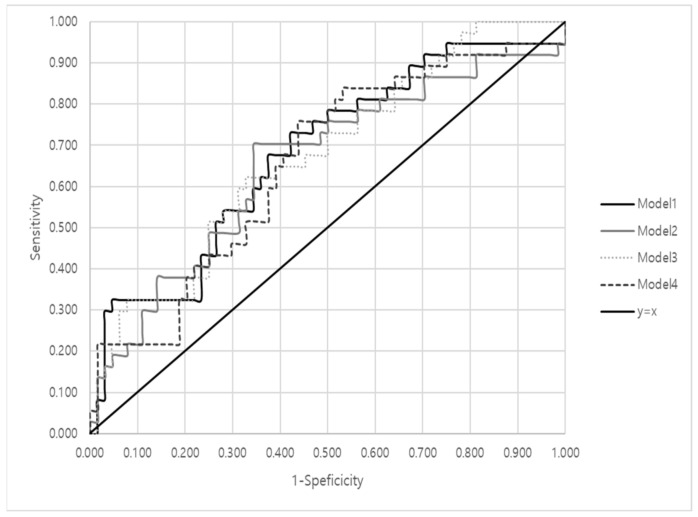
Receiver operating characteristic (ROC) curves of four representative models.

**Table 1 sensors-21-04683-t001:** Time-domain variables for heart rate variability [38].

Variable	Description	Equation
Mean NNI	Mean NN intervals	$\frac{1}{N}\overset{N}{\underset{i=1}{\sum }}RR_{i}$
SDNN	Standard deviation of all NN intervals	$\sqrt{\frac{1}{N}\overset{N}{\underset{i=1}{\sum }}\left(RRi-\bar{RR}\right)}$
SDSD	Standard deviation of differences between adjacent NN intervals	$\sqrt{\frac{1}{N-1}\overset{N-1}{\underset{i=1}{\sum }}{\left(\left|RR_{i}-RR_{i+1}\right|-\bar{RRdif}\right)}^{2}}$
pNNI50	pNN50 count divided by the total number of all NN intervals (%)	$\frac{\sum_{i=1}^{M}\left\{\left|RR_{i+1}-RR_{i}\right|\;>\;50\;\mathrm{ms}\right\}}{N}\times 100$
pNNI20	pNN20 count divided by the total number of all NN intervals (%)	$\frac{\sum_{i=1}^{M}\left\{\left|RR_{i+1}-RR_{i}\right|\;>\;20\;\mathrm{ms}\right\}}{N}\times 100$
RMSSD	The square root of the mean of the sum of the squares of differences between adjacent NN intervals	$\sqrt{\frac{1}{N-1}\overset{N-1}{\underset{i=1}{\sum }}{\left(RR_{i+1}-RR_{i}\right)}^{2}}$
Mean HR	Mean heart rate	$\frac{1}{N}\overset{N}{\underset{i=1}{\sum }}HR_{i}$

**Table 2 sensors-21-04683-t002:** Frequency-domain variables of heart rate variability [38].

Variable	Description	Frequency Range
LF	Power in low-frequency range	0.04–0.15 Hz
HF	Power in high-frequency range	0.15–0.4 Hz
LF/HF ratio	Sympathovagal balance	
LFnu	LF power in normalized units: (LF/(total power − VLF)) × 100	
HFnu	HF power in normalized units: (HF/(total power − VLF)) × 100	
Total Power	The variance of NN intervals over the temporal segment	Approximately ≤ 0.4 Hz
VLF	Power in very low-frequency range	≤0.04 Hz

**Table 3 sensors-21-04683-t003:** The *t*-test results for data from 30 s window size after “killed event”; mean (±standard deviation).

Parameter	Addicted Group	Non-Addicted Group	*p*-Value
MeanNNI	697.72 (±81.42)	672.16 (±82.23)	0.134
SDNN	45.31 (±14.52)	41.2 (±14.53)	0.174
SDSD	28 (±14.05)	23.35 (±12.77)	* 0.093
pNNI50	7.15 (±9.41)	5.21 (±10.9)	0.368
pNNI20	37.53 (±16.67)	29.15 (±17)	** 0.018
RMSSD	28.1 (±14.04)	23.45 (±12.78)	* 0.093
MeanHR	87.5 (±9.86)	90.91 (±10.72)	0.116
LF	1009.35 (±798.74)	679.67 (±596.46)	** 0.020
HF	326.26 (±312.39)	230.59 (±312.38)	0.141
LF/HF ratio	5.16 (±6.8)	6.45 (±10.06)	0.489
LFnu	71.93 (±19.13)	72.41 (±19.2)	0.904
HFnu	28.07 (±19.13)	27.59 (±19.2)	0.904
Total Power	1640.71 (±1084.45)	1260.54 (±970.6)	* 0.072

** *p* < 0.05, * *p* < 0.1.

**Table 4 sensors-21-04683-t004:** The *t*-test results for data from 60 s window size after “killed event”; mean (±standard deviation).

Parameter	Addicted Group	Non-Addicted Group	*p*-Value
MeanNNI	714.36 (±76.27)	690.67 (±80.38)	0.154
SDNN	53.97 (±23.08)	50.21 (±18.52)	0.380
SDSD	29.24 (±13.76)	28.96 (±21.42)	0.943
pNNI50	7.84 (±9.18)	7.14 (±12.55)	0.768
pNNI20	37.88 (±16.86)	32.56 (±17.04)	0.137
RMSSD	29.29 (±13.78)	28.98 (±21.42)	0.939
MeanHR	85.52 (±9.35)	88.8 (±10.15)	0.115
LF	973.06 (±630.72)	753.93 (±536.36)	* 0.071
HF	303.23 (±268.49)	315.82 (±459.75)	0.880
LF/HF ratio	4.21 (±2.75)	4.51 (±4.3)	0.708
LFnu	77.15 (±8.37)	74.02 (±14.85)	0.244
HFnu	22.85 (±8.37)	25.98 (±14.85)	0.244
Total Power	1781.33 (±1129.5)	1672.64 (±1251.71)	0.668
VLF	505.03 (±441.69)	602.89 (±539.92)	0.356

* *p* < 0.1.

**Table 5 sensors-21-04683-t005:** Correlation coefficients among heart rate variability parameters.

	SDNN	SDSD	pNNI50	pNNI20	RMSSD	LF	HF	TOTAL POWER
SDNN								
SDSD	0.684 **							
pNNI50	0.477 **	0.639 **						
pNNI20	0.405 **	0.480 **	0.797 **					
RMSSD	0.686 **	1.000 **	0.639 **	0.480 **				
LF	0.614 **	0.337 **	0.265 **	0.264 **	0.337 **			
HF	0.483 **	0.608 **	0.402 **	0.267 **	0.607 **	0.545 **		
Total Power	0.731 **	0.460 **	0.320 **	0.275 **	0.461 **	0.958 **	0.680 **	

** Correlation is significant at the 0.01 level (both sides).

**Table 6 sensors-21-04683-t006:** The eigenvalues from factor analysis.

	Factor 1	Factor 2
SDNN	0.664	0.593
SDSD	0.860	0.064
pNNI50	0.838	−0.075
pNNI20	0.766	−0.081
RMSSD	0.860	0.066
LF	0.450	0.715
HF	0.686	0.101
Total Power	0.559	0.705

**Table 7 sensors-21-04683-t007:** Summary results of four logistic regression models with the highest accuracy.

Model No.	Parameter	Model Equation	Sensitivity	Specificity	Accuracy (%)
Model 1	pNNI20 RMSSD LF	1/(1 + exp (−(−1.705 + 0.048 × pNNI20 − 0.035 × RMSSD + 0.0005 × LF)))	0.324	0.906	69.3
Model 2	ln(pNNI20) (RMSSD)2 ln(LF)	1/(1 + exp (−(−9.911 + 2.309 × ln(pNNI20) + 0.0003 × (RMSSD)^2^ + 0.272 × ln(LF))))	0.400	0.828	67.7
Model 3	pNNI20 ln(RMSSD) LF	1/(1 + exp (−(1.232 + 0.054 × pNNI20 − 1.292 × ln(RMSSD) + 0.0006 × LF)))	0.324	0.922	70.3
Model 4	ln(pNNI20) ln(RMSSD) (LF)2	1/(1 + exp (−(−5.5417 + 2.505 × ln(pNNI20) − 1.271 × ln(RMSSD) + 0.0000002 × LF^2^)))	0.891	0.343	69.7

## Data Availability

Not applicable.

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
