# Peer review of "Identification of Video Game Addiction Using Heart-Rate Variability Parameters"

_sensors, 2021, doi:10.3390/s21144683_

Round 1

Reviewer 1 Report

The purpose of this study is to extract quantitative heart rate variability (HRV) parameters that characterize the particular stress response of game addicts. A battle game, “League of Legends”, was used as the stimulus for 23 experiment participants. Twelve subjects - healthy controls. The authors investigated 14 standard HRV parameters from the electrocardiogram. The authors selected particular events for analysis: “killed event,” when a player’s character was killed by an opponent, and a “killing event,” when the player killed an opponent. The data collection window sizes for these events were 30 s, 60 s, 90 s, and 120 s, respectively. The results show poor discrimination power of the proposed methodology to game addicts and non-addicts - AUC = 0.677.

Investigated problem is very important and relevant; however, the purpose of the study is not achieved. Lack of methodological novelty (all the proposed HRV parameters are standard), non-acceptable discriminating power of the used methodology warrant further work in obtaining more data and elaborating better markers to characterize the particular stress response of game addicts.

Other comments:

1. The authors did not state any novelty of the study in the Introduction.

2. The t-test was used to test the parameters in differentiating two groups game addicts and non-addicts. However, t-test is suitable for Gaussian distributed data. Have authors checked the normality of their data?

3. The authors were using frequency-domain parameters LF, HF, LF/HF ratio, LFnu, HFnu, total power, and VLF. I have a doubt about the usage validity of these parameters, especially in the shortest windows 30s and 60s. VLF parameter takes into account frequencies <0.04Hz, i.e., with the period of 25s. This means hardly one period of the signal fits the shortest window of the 30s. How were frequency-domain parameters calculated: by using non-parametric methods (FFT, Welch) or parametric?

Author Response

Reviewer 1

The purpose of this study is to extract quantitative heart rate variability (HRV) parameters that characterize the particular stress response of game addicts. A battle game, “League of Legends”, was used as the stimulus for 23 experiment participants. Twelve subjects - healthy controls. The authors investigated 14 standard HRV parameters from the electrocardiogram. The authors selected particular events for analysis: “killed event,” when a player’s character was killed by an opponent, and a “killing event,” when the player killed an opponent. The data collection window sizes for these events were 30 s, 60 s, 90 s, and 120 s, respectively. The results show poor discrimination power of the proposed methodology to game addicts and non-addicts - AUC = 0.677.

Investigated problem is very important and relevant; however, the purpose of the study is not achieved. Lack of methodological novelty (all the proposed HRV parameters are standard), non-acceptable discriminating power of the used methodology warrant further work in obtaining more data and elaborating better markers to characterize the particular stress response of game addicts.

Other comments:

  1. The authors did not state any novelty of the study in the Introduction.

HRV has been used in many researches; however, it is not yet established which parameter or combination of parameters could be the best discriminator to detect game addiction. In particular, the current study examined the all existing HRV parameters used in frequency domain and time series in order to find the most effective set of parameters via bottom-up exhaustive process, that was very time consuming and data intensive. I think it is a novel effort to discover new set of parameters with given HRV parameters. This was described in line 72-81 of. Moreover, this data-driven process was also intended to future use of AI algorithm including machine learning.

  1. The t-test was used to test the parameters in differentiating two groups game addicts and non-addicts. However, t-test is suitable for Gaussian distributed data. Have authors checked the normality of their data?

Yes, the normality of the data was tested in this study.  Generally speaking, the bio-signal from human data was normally distributed by nature as long as the enough number of samples were collected.  The most samples in this study showed high normality values. However, there were cases that even the legally (without any experimental error) collected data in this study consisted of statistical outliers to be eliminated, that could lower the normality value. This phenomenon is often observed when dealing with bio-signal with great variability like EEG, EMG, ECG etc.  In such case, if the experimenter decided to treat the extreme data as outliers, a potentially significant information can be lost at the cost of having high normality value of sample. Otherwise, the experimenter should include the extreme data as legal and useful data from the subject at the cost of having insufficient normality value in particular sample. This is the limitation of using only t-test to discriminate the bio-signal with great variability among heathy and addicted subjects in any studies. Therefore, this study supplemented the correlation analysis and factor analysis to compensate the potentially insufficient statistical power of t-test in order to improve the sensitivity of the outcomes.

  1. The authors were using frequency-domain parameters LF, HF, LF/HF ratio, LFnu, HFnu, total power, and VLF. I have a doubt about the usage validity of these parameters, especially in the shortest windows 30s and 60s. VLF parameter takes into account frequencies <0.04Hz, i.e., with the period of 25s. This means hardly one period of the signal fits the shortest window of the 30s. How were frequency-domain parameters calculated: by using non-parametric methods (FFT, Welch) or parametric?

Since the authors examined the window size up to 120 seconds, it was legitimate way to examine all the parameter performance with low frequency nature including the VLF.  And, the author used the elimination process of inefficient parameters by testing parameters in various window sizes.  During the process, the efficient parameters were found in 30 second window and naturally other parameters were dropped out. 

In this study, the frequency conversion was performed by using FFT.

Reviewer 2 Report

Diagnosis of psycchiatric conditions are highly subjective, therefore any attampt to identify objective, measureable variable suggesting a pathological state should be welcomed. This study aimes to identify HRV parameter for diagnosis of game addicts. Althought previous studies identified  such "biomarkers" of addiction on EEG, but ECG is far easier to obtain, therefore it can be used more widely. The paper is well written, easy to follow and understand. Some minor questions remained to be elucidated.

L84 How were the participants recruited/selected? What was their gender, major?

L92 How many students were discarded?

L154 I am not convinced that t-test is the best method for comparison many, connected data (see dependent variables l117)

Author Response

Reviewer 2

Diagnosis of psycchiatric conditions are highly subjective, therefore any attampt to identify objective, measureable variable suggesting a pathological state should be welcomed. This study aimes to identify HRV parameter for diagnosis of game addicts. Althought previous studies identified such "biomarkers" of addiction on EEG, but ECG is far easier to obtain, therefore it can be used more widely. The paper is well written, easy to follow and understand. Some minor questions remained to be elucidated.

L84 How were the participants recruited/selected? What was their gender, major?

L92 How many students were discarded?

14 addicted subjects and 14 non-addicted subjects were selected. 3 addicted subject and 2 non-addicted subjects were discarded due to the technical error in measurement system. Controlling the compounding effect of gender in this study, only male participants were recruited in this study. It is added in line 92-96.

L154 I am not convinced that t-test is the best method for comparison many, connected data (see dependent variables l117)

T-test should not be a single method to discriminate two groups.  However, without basic t-test, the author may miss the basic difference of two groups based on the mean and std. difference. That is why this study combined the correlation analysis and factor analysis together to extract the best parameters, and finally sorted out the parameters with cohering outcomes in three different analyses.

Reviewer 3 Report

  1. The purpose of your paper is to use an electrocardiogram (ECG) data to determine heart rate variability (HRV) parameters that can quantitatively characterize game addiction and to extract them from existing indicators. As a result, you concluded the PNNI20 and LF parameters using 3 different statistical methods.
  2. Internet games relieve the stress associated with daily life and feelings of loneliness. However, the higher the level of stress, the more likely it is to become game addiction, and game addiction causes various psychological problems and stress reactions. These are clear from previous studies and have already been pointed out by you. In short, in the first place, what I mean is that game addictors are stressful and physical / mental unhealthy ?
  3. Or the stress of game addiction different from the stress of daily life, and it has possibility that some intervention will reduce stress? (e.g., shortening the game time subjectively / objectively reduce stress index, etc.)
  4. You use a questionnaire to classify subjects into healthy and addicted, but what is criteria of classification? Is it like recording the time of the game? It should be noted that the long playing game time is not exclusively addiction.
  5. Dependence is the repetitive use of drugs and chemicals with or without physical dependence, and intoxication is different from addiction. (e.g., food poisoning caused by rotten fish is unlikely to become a dependence.) Addiction is a condition in which substance use is repeated, the amount used increases, and when it becomes unusable, serious symptoms appear, the uncontrollable urge to use increases. And physical / mental deterioration. It is well known that it can lead to serious mental illness. First, you should state in your paper the definitions of "Video game addiction" and "gaming disorder (internet gaming disorder)". And the classification of subjects should be done in detail. Although the number of patients with gaming disorders is not so much, it would be a useful study for society if one healthy person onset gaming disorders for some reason and evaluate them objectively. I think your point of view is good.
  6. Your research has a great point of view in society, the experimental protocol is generally good, and there are no problems with analysis. However, there are many corrections in your paper, and it seems difficult counterargument  to for 2. After describing the limitation of this research, recommend postings other than this journal.

Author Response

Thank you for your comments.

Reviewer 4 Report

This paper present results of an experiment correctly designed, with scientific interest but with an unpromising result.

I have not major comments related with the methodology or the discussion, because they are correct.

The main drawback of this paper is that data are not available to the research community and so results are note verifiable. Why you say that "data available statement" is "not aplicable"?

Minor comments:

-Table 4. HF decrease in addicted group while it increases accoridng to table 3. Also LF increase, HF decrease but LF/HF decrease.  Please check is values showed in table 4 are correct.

- ref 24. The title is not correct

- ref 37. Please update the access date.

Author Response

Reviewer 4

This paper present results of an experiment correctly designed, with scientific interest but with an unpromising result.

I have not major comments related with the methodology or the discussion, because they are correct.

The main drawback of this paper is that data are not available to the research community and so results are note verifiable. Why you say that "data available statement" is "not applicable"?

The amount of raw data in this experiment is massive and the data contains the personal information of addicted subjects, and they did not consent to release their information. Author could only release the statistically summarized form for academic community to share the discovered knowledge. Any researchers could verify the outcomes by following the methodology this study has reported in detail.

Minor comments:

-Table 4. HF decrease in addicted group while it increases according to table 3. Also LF increase, HF decrease but LF/HF decrease.  Please check is values showed in table 4 are correct.

We checked the table 3 and 4, found no errors. Due to the large standard deviation, the average value was affected and seemed to be a bit fluctuating, but the measured value was correctly reported.

- ref 24. The title is not correct

It is corrected in line 381 as follows:

Traina, M.; Cataldo, A.; Galullo, F.; Russo, G., Effects of anxiety due to mental stress on heart rate variability in healthy subjects. Minerva Psichiatrica 2011, 227, 31.

- ref 37. Please update the access date.

It is corrected in line 408 as follows:

Most Popular Core PC Games. Available online: https://newzoo.com/insights/rankings/top-20-core-pc-games (accessed on April, 2021).

Round 2

Reviewer 1 Report

1. The revised version shows that the authors mostly ignored the reviewers' comments.   

2. Please, include, which normality test did you use? Kolmogorov–Smirnov test, Anderson–Darling test? https://en.wikipedia.org/wiki/Normality_test

3. It is illegal to calculate VLF parameter in the 30s long window. Please remove it from Table 3.  

4. The authors state "the author used the elimination process of inefficient parameters by testing parameters in various window sizes." I did not find any description about the "elimination process" in the manuscript.

5. I suggest removing Fig 1 as it is too trivial to show a generic ECG beat in scientific publication.   

Author Response

We thank you for your time and consideration on our submission.

Reviewer 3 Report

none

Author Response

(The authors gave the same response as above.)
